# Phosphorus Plays Key Roles in Regulating Plants’ Physiological Responses to Abiotic Stresses

**DOI:** 10.3390/plants12152861

**Published:** 2023-08-03

**Authors:** Fahad Khan, Abu Bakar Siddique, Sergey Shabala, Meixue Zhou, Chenchen Zhao

**Affiliations:** 1Tasmanian Institute of Agriculture, University of Tasmania, Launceston, TAS 7250, Australia; fahad.khan@utas.edu.au (F.K.); abubakar.siddique@utas.edu.au (A.B.S.); meixue.zhou@utas.edu.au (M.Z.); 2School of Biological Science, University of Western Australia, Crawley, WA 6009, Australia; sergey.shabala@uwa.edu.au; 3International Research Centre for Environmental Membrane Biology, Foshan University, Foshan 528000, China

**Keywords:** abiotic stresses, physiological responses, stomatal functioning, low phosphorus, phosphorus deficiency

## Abstract

Phosphorus (P), an essential macronutrient, plays a pivotal role in the growth and development of plants. However, the limited availability of phosphorus in soil presents significant challenges for crop productivity, especially when plants are subjected to abiotic stresses such as drought, salinity and extreme temperatures. Unraveling the intricate mechanisms through which phosphorus participates in the physiological responses of plants to abiotic stresses is essential to ensure the sustainability of agricultural production systems. This review aims to analyze the influence of phosphorus supply on various aspects of plant growth and plant development under hostile environmental conditions, with a special emphasis on stomatal development and operation. Furthermore, we discuss recently discovered genes associated with P-dependent stress regulation and evaluate the feasibility of implementing P-based agricultural practices to mitigate the adverse effects of abiotic stress. Our objective is to provide molecular and physiological insights into the role of P in regulating plants’ tolerance to abiotic stresses, underscoring the significance of efficient P use strategies for agricultural sustainability. The potential benefits and limitations of P-based strategies and future research directions are also discussed.

## 1. Introduction

The projected global population of 9.9 billion by 2050 necessitates an over 70% increase in food production [1]. Therefore, the effective utilization of resources to improve agricultural production is extremely crucial. A substantial portion of soils worldwide lacks phosphorus (P), resulting in a severe limitation of crop yields and posing a substantial risk to global food security [2]. Phosphorus is found in abundance in the lithosphere but the form used by plants, inorganic orthophosphate (Pi), is insoluble and diffuses slowly in soils [3]. This leads to P deficiency in agricultural lands and ecosystems. The availability of P is also affected by soil microbiota, which can either compete with plants for P or establish beneficial relationships, such as mycorrhizae, to increase P acquisition efficiency [4]. The inefficiency of P fertilizers is due to their properties, with only 15–25% being taken up by plants while the remaining amount is leached, causing soil degradation and water eutrophication [5,6]. In contrast to nitrogen, which is abundantly available in the atmosphere, rock phosphate (rock P) has a limited supply. If the current consumption rates persist, there is a potential for a phosphorus fertilizer shortage crisis by the end of the current century [7]. In light of the aforementioned facts, it is hardly surprising that studies related to phosphorus have received increasing attention as depicted by the number of publications growing each year from 2000 to 2022 (Figure 1). As agriculture is moving toward more sustainable practices with a reduced reliance on synthetic fertilizers, it is essential to unravel the molecular and physiological mechanisms that underlie plant adaptations to P limitation. Of specific interest is the impact of P availability on stomatal development and operation as this knowledge could provide insights into how plants allocate resources to withstand nutrient limitation and maintain water use efficiency, ultimately leading to the production of crop varieties that are more resilient and productive under low-nutrient conditions.

Phosphorus plays a fundamental role in regulating physiological responses and enhancing abiotic stress tolerance in plants, such as heat, salinity, drought, waterlogging, high CO_2_ and heavy metal toxicity [8,9]. Plants possess the capability to sense and respond to variations in phosphorus availability through specific signaling pathways, root architecture modulation and stomatal morphology adjustments [10,11]. Moreover, by modulating their phosphorus metabolism, plants can tolerate diverse abiotic stresses including heat, drought, salinity and heavy metal toxicity, where stomatal responses are particularly relevant [12,13,14,15]. Understanding the mechanisms by which plants sense and respond to phosphorus availability is crucial for developing strategies that enhance crop productivity and stress tolerance. Phosphorus is also an essential nutrient for diverse metabolic and physiological processes encompassing energy metabolism, cell division, DNA synthesis and phospholipid biosynthesis, primarily as phosphate (Pi) or Pi esters [16]. Insufficient Pi in the soil adversely affects fruit production, quality traits during vegetative growth and root development, ultimately resulting in reduced crop yields [17,18,19].

Providing soluble P to plants during their growth cycle is a commonly recommended approach. This can be achieved by applying phosphorus pentoxide (P_2_O_5_) as a fertilizer [20]. Depending on the specific crop species, soil properties and agricultural practices, fertilizers may be applied at rates of up to 120 kg P ha^−1^ [21,22]. However, this traditional method of P supply has accounted for excessive P accumulation in soils in recent decades. With proper management, interventions and cropping systems, this surplus P can become a valuable source of P legacy [23]. Given the growing concerns of climate change and the detrimental effects of multiple crop stressors, it is imperative to adopt innovative technologies that enhance P utilization efficiency and management. These innovative approaches include the utilization of microbes for P solubilization [24,25], the implementation of partially activated P [26], the development of slow/controlled release P fertilizers [27,28], the application of nanotechnology [29] and the formulation of foliar fertilizers [30]. Even though these innovative approaches alleviate the effects of low P, given the high cost and threats to the environment, more efficient and low-cost methods should be pursued. To satisfy this purpose, understanding the physiological and molecular mechanisms of plants’ response to low P becomes more necessary.

The intricate relationship between the soil, rhizosphere and plants plays a pivotal role in determining the availability of P in plants. Here, “rhizosphere” refers to the region nearby the root system where active interactions between plant roots, the soil environment and soil microbes take place. The spatial and temporal variations in P distribution and dynamics within the soil necessitate a well-adapted root architecture that allocates more roots to P-rich areas, facilitating efficient utilization of P resources [31]. Moreover, the root architecture demonstrates functional coordination by exuding carboxylates, protons and phosphatases, which aids in P mobilization and acquisition. Plants adapt their root morphology and physiology to P-limited environments to effectively match the heterogeneous supply and distribution of P in the soil, ultimately enhancing the spatial availability and bioavailability of P. The integration of P dynamics across the soil-to-plant continuum through the rhizosphere provides a comprehensive understanding of P behavior and efficient acquisition in conjunction with plant adaptive strategies [32]. In recent decades, substantial progress has been made in unraveling the intricate processes related to soil P transformation, mobilization, acquisition and plant responses to P deficiency [33]. However, certain aspects of the overall P dynamics within the soil/rhizosphere–plant system remain understood only partially. These include the regulation of P acquisition and P starvation rescue mechanisms in plants, the intricate coordination of root morphology, physiological and biochemical responses under varying P availability, and the plant’s perception of heterogeneous P supply within the soil. Further exploration and investigation are required to deepen our understanding of these crucial aspects in order to optimize P management in agricultural systems and enhance crop productivity.

Gaining a profound and all-encompassing understanding of the intricate phosphorus dynamics spanning across the soil-to-plant continuum has paramount significance, as it enables the optimization of phosphorus management strategies, the enhancement of phosphorus-use efficiency, the reduction of reliance on chemical phosphorus fertilizers, the exploitation of rhizosphere processes’ potential for efficient mobilization and the facilitation of plant uptake of soil phosphorus, including phosphorus recycling from waste and manure sources [11]. The dynamics of phosphorus within the soil–plant system are profoundly influenced by the integrated and synergistic effects of phosphorus transformation, availability and utilization, which are intricately governed by the intricate interplay of plant, soil and rhizosphere processes. This review elucidates the multifaceted phosphorus dynamics in the soil/rhizosphere–plant continuum, with a specific emphasis on the key factors that influence phosphorus availability in the soil and rhizosphere, the intricate mechanisms of phosphorus mobilization, the intricate processes of plant uptake and the utilization of phosphorus resources (Figure 2). Among the topics covered are (1) the impact of P availability on the physiological, molecular and metabolic characteristics of plants grown under adverse environmental conditions; (2) the potential of P supplementation in improving plant tolerance to various environmental stresses; (3) newly discovered genes related to P-dependent stress regulation; and (4) the feasibility of implementing P-based agricultural practices to alleviate abiotic stress.

## 2. Physiological Adaptations to Low P and Regulatory Networks of Phosphorus in Plant Growth

Phosphorus is typically taken up by plants as inorganic phosphate (Pi) from the soil. Once transported into the plant, Pi is converted to organic forms such as adenosine triphosphate (ATP) and adenosine diphosphate (ADP), which are used as energy carriers. Pi is also used to form other important organic compounds like nucleic acids, phospholipids and inorganic polyphosphates (polyP), thus conferring plant growth and physiological and molecular adaptations [34]. Figure 3 provides an outlook of the functions related to phosphorus availability in the shoots, roots and leaves of plants. The physiological and molecular insights of phosphorus homeostasis are discussed below:

### 2.1. Growth Retardation and Yield Loss under Low P

Phosphorus is essential for the synthesis and activation of enzymes involved in nutrient acquisition and transport and its deficiency can lead to nutrient imbalances and toxicity symptoms. Insufficient P affects the uptake of other nutrients like nitrogen (N), potassium (K) and calcium (Ca) by plants, altering their architecture, leading to things such as reduced stem elongation and the production of shorter and thicker stems [35]. P deficiency also influences biomass allocation, with a drop in above-ground biomass and a rise in root biomass. Maize biomass peaks at a shoot P concentration of 2.7 mg g^−1^, with root morphology showing an increasing trend at lower P concentrations but decreasing at extremely low levels. Conversely, increasing the shoot P concentration results in decreased root morphology but enhanced physiological responses, even at high P concentrations, highlighting the importance of the root structure in P uptake and utilization [36]. P deficiency also affects leaf morphology and photosynthesis. For instance, certain rice genotypes exhibit better adaptation to low P conditions with higher biomass, increased photosynthetic rates and enhanced antioxidant enzyme activities compared to other varieties [37]. Similar variations are observed among potato cultivars, with P-efficient cultivars demonstrating higher biomass, tuber yield and P uptake efficiency under low P conditions, emphasizing the significance of selecting P-efficient cultivars for improved P efficiency and tuber quality [38]. P deficiency delays flowering and fruiting, reduces seed production and alters plant architecture. *Arabidopsis thaliana* responds to low P availability with phenological delay, allowing for increased P acquisition and utilization, resulting in higher reproductive biomass production [39]. Low-phytate soybean lines, despite having reduced phytate P concentration, exhibit similar growth, photosynthesis, nitrogen fixation, seed yield and seed quality compared to normal phytate cultivars, suggesting their potential for improving nutrient utilization [40]. In wheat crops, P deficiency primarily affects above-ground biomass, radiation interception, grain yield and spike biomass, leading to reductions in the grain number per square meter [41]. Additionally, P deficiency in soft red winter wheat decreases the leaf appearance rate, final number of leaves and leaf area per plant and hampers the plant’s ability to cope with mild water stress [42]. Overall, P deficiency not only affects nutrient acquisition and transport but also induces changes in plant morphology, biomass allocation, leaf morphology and photosynthesis, underscoring its importance in various physiological processes and overall plant growth and development.

### 2.2. Root Architecture and Exudation

Plants have developed adaptive mechanisms to cope with P deficiency by modifying their root morphology and architecture, which improves P uptake efficiency. When faced with low-P conditions, plants exhibit an increased production of longer root hairs and greater lateral root growth to enhance P absorption [43]. P deficiency negatively affects leaf expansion, biomass accumulation and root development due to the impact on the plant’s carbon budget. The initial increase in carbohydrate availability for root growth is followed by a decrease in leaf area, limiting light interception and subsequently reducing root growth [44].

The hormone auxin plays a crucial role in regulating the root system architecture under low-P conditions. It promotes the growth of lateral roots and enhances root hair development, thereby adding to the surface area available for P uptake [45]. Other hormones, including cytokinins, gibberellins and abscisic acid, have also been found to be important in regulating plant responses to P deficiency [46,47,48,49]. Phosphorus deficiency stimulates the formation of specialized root structures called cluster roots, characterized by an abundance of root hairs. These root hairs effectively absorb P from the soil and their formation is regulated by specific genes [50,51]. Plants optimize P uptake by modulating their root architecture and facilitating the acidification of the rhizosphere through increased proton pump activity. Acidification enhances P solubility in the soil and facilitates its uptake by roots [52]. Root exudation stands as another significant strategy employed by plants to cope with P deficiency. This process involves the active release of organic compounds from the roots into the rhizosphere, subsequently altering the soil’s properties and facilitating P acquisition [53]. Studies have demonstrated a notable rise in the exudation of citrate, malate and oxalate under low P, resulting in a significant improvement in P uptake efficiency [54]. Selecting cultivars with resistance to both low P and aluminum toxicity may be advantageous, as organic acid formation and exudation under these stresses occur at different stages [55]. Root exudates also engage in interactions with microorganisms dwelling in the rhizosphere, fostering the growth of beneficial microorganisms such as arbuscular mycorrhizal fungi (AMF) that actively contribute to P uptake [56]. These symbiotic fungi establish intricate associations with plant roots, facilitating the transfer of P from the soil in exchange for carbon compounds derived from photosynthesis [57]. The root exudates serve as a valuable carbon source, stimulating the growth and colonization of mycorrhizal fungi on plant roots. Furthermore, root exudates play a pivotal role in shaping the microbial community within the rhizosphere, favoring the proliferation of phosphate-solubilizing bacteria that enhance P availability by promoting mineral dissolution [58,59].

In summary, plants employ a diverse array of strategies to enhance P uptake efficiency, including modifications to their root morphology, the exudation of organic acids and the establishment of symbiotic associations with mycorrhizal fungi. Root exudates also exert a critical influence on the microbial community within the rhizosphere, fostering beneficial interactions that further contribute to increased P availability for plants.

### 2.3. Molecular Responses to Low P

Significant progress has been made in understanding the genetic response to low P stress, with four distinct gene groups involved in various aspects of the plant’s adaptation being identified [4]. One of these groups is responsible for sensing low P signals, encompassing both local phosphate sensing in the primary root of *Arabidopsis thaliana* [60] and long-distance phosphate signaling mediated by transcription factors such as PHR1 and PHL1 [61]. Another group of genes regulates phosphate distribution and maintains phosphorus homeostasis, including SPX family members and PHO1 with an SPX domain in Arabidopsis [62]. The remaining two gene groups primarily focus on phosphate acquisition and translocation. For example, GmACP1 encodes an acid phosphatase that, when overexpressed in roots, enhances phosphate absorption from the nutrient solution [63]. Additionally, the PHT1 family, comprising 15 PHT1 genes that encode phosphate transporters involved in both root phosphate uptake and intercellular phosphate transport [64,65]. The root system plays a key role in phosphate acquisition and transportation, as evidenced by the changes in root morphology under P-deficient conditions. Under low P, maize exhibits increased lateral root number and length [66], while low-P tolerant accessions of barley and Arabidopsis display longer and more abundant root hairs [67,68]. Similarly, overexpression of the low-phosphorus-related transcription factor OsMYB2P-1 in rice leads to extended primary roots [69]. Notably, QTL mapping studies have revealed QTL linked to seed phosphorus content [70], phosphorus-efficient root traits [71] and plant tolerance to low phosphorus stress based on flower and pod abscission rate in soybeans [72].

Low P triggers complex and diverse responses in plants. Phosphate deprivation strongly influences phytohormone signaling pathways, impacting root growth in Arabidopsis and *Zea mays* [73,74]. Long-term phosphorus deficiency leads to anthocyanin accumulation, indicating changes in secondary metabolites [73]. Transcriptome analyses on Arabidopsis, rice and *Zea mays* have uncovered numerous genes and signaling pathways associated with low-P stress [73,75]. Similarly, gene expression studies on *Zea mays* revealed the differential expression of genes involved in the phenylpropanoid pathway during P starvation [74]. Zeng et al. employed high-throughput sequencing to identify phosphate-deficiency-responsive genes in the roots of soybean plants [76]. The identification of P-efficient plant materials and the exploration of high-phosphorus-efficient genes are crucial endeavors aimed at mitigating the adverse impacts of phosphorus deficiency on plant growth. By focusing on these aspects, we can effectively address the challenges associated with phosphorus limitation, enhancing the overall productivity and sustainability of agricultural systems.

### 2.4. Hormonal Responses to Low P

Cytokinins (CKs) play a crucial role in the regulation of plant responses to environmental stresses and interact extensively with abscisic acid (ABA) signaling pathways. Studies have indicated that CKs may act as negative regulators in P starvation responses (PSRs). For instance, exogenous CKs were found to repress the expression of PSR genes in Arabidopsis under low-P conditions [77]. CKs are also involved in the regulation of lateral root development during P starvation [78]. Additionally, CK signaling pathways affect P signaling, and mutations in CK receptors can influence PSRs [79]. Ethylene, another plant hormone, is also important in the regulation of PSRs, especially concerning root hair development [80]. Ethylene treatment affects plant sensitivity to P starvation, and mutations in ethylene-related genes can lead to hypersensitive responses to low P conditions [81]. Strigolactones (SLs) are associated with P deficiency, with SL exudation being induced by P starvation. SLs may also be involved in root response to P starvation and interact with AUX signaling for lateral root development and root hair density [82]. Lastly, gibberellin (GA) has been implicated in PSRs, with DELLA domain proteins, critical GA signal mediators, playing a role in regulating anthocyanin accumulation and root architecture changes under P starvation conditions [83]. Functional analysis of MYB62 suggests its involvement in PSRs through changes in GA metabolism and signaling [84].

Plant responses to both low and excess phosphorus involve complex interactions with various hormone-signaling pathways. CKs, ABA, ethylene, SLs and GA all contribute to the regulation of PSRs, influencing processes such as gene expression, lateral root development and shoot branching. Understanding the intricate crosstalk between these hormone-signaling pathways and phosphorus responses is essential for unraveling the molecular mechanisms behind plant adaptations to varying phosphorus conditions.

### 2.5. Phosphorus Homeostasis

Plants maintain phosphorus (P) homeostasis through a series of regulated processes involving P uptake, utilization and remobilization when faced with P-deficient conditions. The signaling pathways responsible for P uptake include the PHT1 family, consisting of high-affinity phosphate transporters, and the P starvation response (PSR) pathway, which involves transcription factors like PHOSPHATE STARVATION RESPONSE 1 (PHR1) and PHR1-LIKE 1 (PHL1) [85,86]. The PSR pathway activates the expression of genes involved in P uptake, transport and utilization, such as phosphate transporters (PHTs), phosphatases and purple acid phosphatases (PAPs), thus enhancing the efficiency of P acquisition and utilization [87,88].

The PHT1 family, which is predominantly expressed in the roots of plants, assumes a pivotal role in phosphorus uptake by facilitating the transportation of inorganic phosphate (Pi) from the soil. This family consists of multiple isoforms of PHT1 transporters, each exhibiting diverse affinities for phosphorus. Certain isoforms demonstrate remarkable efficiency in phosphorus uptake, particularly under conditions of limited phosphorus availability [86]. Other transporters, such as PHOSPHATE TRANSPORTER TRAFFIC FACILITATOR1 (PHF1) and PHOSPHATE TRANSPORTER1 (PHT1;8), have also been implicated in P uptake during P deficiency [89,90].

When P concentrations in the soil are low, the PSR pathway is activated. Following phosphorylation and subsequent translocation to the nucleus, transcription factors PHR1 and PHL1 play a pivotal role in orchestrating the activation of genes associated with phosphorus uptake, transport and utilization [91]. Additionally, other transcription factors, including WRKY74, WRKY75, MYB2, MYB4 and ARF16, have been associated with the PSR pathway [92,93]. By exerting regulatory control over gene expression, these transcription factors actively govern the transcriptional activity of genes encoding crucial components involved in phosphorus uptake, transport and utilization. This includes a wide array of transporters, enzymes and proteins, such as PHTs (phosphate transporters), PAPs (purple acid phosphatases) and ribonucleases. Through their influence, these transcription factors work in synergy to enhance the overall efficiency of phosphorus acquisition and utilization in plants.

In addition to the PSR (phosphate starvation response) pathway, there exist multiple other signaling pathways that actively contribute to the intricate regulation of phosphorus homeostasis in plants. These pathways encompass the mitogen-activated protein kinase (MAPK) pathway, calcium-dependent protein kinase (CDPK) pathway and target of rapamycin (TOR) pathway. The MAPK pathway exhibits responsiveness to diverse biotic and abiotic stresses, including phosphorus deficiency, thus modulating gene expression and metabolic processes in plants [92,94]. The CDPK pathway, activated by fluctuations in intracellular calcium levels, governs a plethora of processes, ranging from stress responses to developmental events [95]. As for the TOR pathway, it represents a highly conserved signaling pathway that coordinates cellular growth and metabolism in response to nutrient availability and stress cues [96]. It has been demonstrated to exert regulatory control over phosphorus homeostasis in plants, encompassing aspects such as phosphorus uptake, utilization and remobilization [97].

To maintain phosphorus homeostasis, plants rely on the dynamic process of P remobilization from older tissues to younger tissues, particularly during reproductive growth stages observed in crops like maize and rice [98]. P remobilization encompasses various intricate mechanisms, including the degradation of phospholipids and nucleic acids, the mobilization of P from vacuoles and senescing tissues and other associated processes [99,100]. The efficiency of P remobilization is regulated by factors such as gene expression involved in P remobilization, enzymatic activity responsible for P degradation and mobilization and the availability of energy sources to support the process [101].

In summary, plants employ a sophisticated network of signaling pathways, including the phosphate starvation response (PSR), mitogen-activated protein kinase (MAPK), calcium-dependent protein kinase (CDPK) and target of rapamycin (TOR) pathways, to meticulously regulate phosphorus homeostasis. These pathways orchestrate P uptake, transport and utilization in a coordinated manner. Additionally, plants ensure a constant supply of phosphorus by effectively remobilizing P from older tissues to younger tissues. The regulation of this process involves gene expression, enzymatic activity and the availability of energy sources. Therefore, gaining a comprehensive understanding of these intricate signaling pathways is imperative for the development of improved plant varieties with enhanced phosphorus efficiency under conditions of limited phosphorus availability.

## 3. Stomatal Activity and Photosynthetic Efficiency in Response to P Deficiency

Sustaining an optimal concentration of phosphorus is vital for promoting healthy plant growth and development. On one hand, excessive accumulation of phosphorus can prove toxic, while on the other hand, inadequate supply can trigger a decline in photosynthesis and hinder various physiological processes. The response of plants to phosphorus supply is contingent upon the species and is influenced by the concentration of this essential nutrient. In the context of phosphorus deficiency, photosynthesis, a fundamental process for plant growth, is curtailed, with the underlying mechanisms being either stomatal or biochemical in nature.

### 3.1. Stomatal Responses to Low P

Stomatal activity stands as a pivotal process regulating the exchange of gases between plants and their surroundings. Stomatal density, size and aperture serve as crucial factors governing the rate of gas exchange. When phosphorus is lacking, stomatal density can decrease, accompanied by reductions in the stomatal size and aperture. This decline in the density and size can yield a decrease in the uptake rate of CO_2_, ultimately leading to diminished rates of photosynthesis [102]. Furthermore, phosphorus deficiency can impact stomatal conductance, which quantifies the rate of water vapor diffusion through stomata. Numerous factors, including the concentration of abscisic acid (ABA), a plant hormone that promotes stomatal closure, govern stomatal conductance. Insufficient phosphorus can elevate the concentration of ABA, causing reduced stomatal conductance and heightened water loss through transpiration [103,104].

Chickpea plants receiving adequate phosphorus and water exhibited enhanced stomatal density and conductance, consequently boosting photosynthetic efficiency and nutrient uptake [105]. In barley seedlings, phosphorus deficiency exerted a more pronounced influence on growth, water relations and gas exchange parameters compared to salinity stress. Furthermore, when exposed to both salinity and phosphorus deficiency, the plants responded similarly to those grown under phosphorus deficiency alone, indicating the predominant influence of phosphorus constraint [106]. Deficiencies in phosphorus and potassium in tea plants caused a reduction in stomatal aperture and an increase in leaf water potential, leading to decreased transpiration. While phosphorus deficiency reduced stomatal density, potassium deficiency increased it. Moreover, potassium deficiency amplified the relative sensitivity of transpiration to water stress, whereas phosphorus deficiency did not [107]. In cotton plants, phosphorus deficiency induced greater sensitivity to water stress, resulting in stomatal closure at higher leaf water potentials compared to plants supplied with adequate phosphorus. P-deficient plants also accumulated higher levels of abscisic acid (ABA) and displayed increased sensitivity to ABA under non-stressed conditions, which was partially mitigated by kinetin [108]. Soybean plants experiencing phosphorus deprivation exhibited stomatal closure, which was intensified by girdling. This closure coincided with elevated foliar ABA accumulation and increased expression of ABA biosynthesis genes [109]. While phosphorus deficiency reduced stomatal conductance in rice plants, it did not limit leaf photosynthetic rates. Instead, it prompted greater carbohydrate allocation to the roots, contributing to the enhanced low-phosphorus tolerance [110]. In phosphate-starved tomato plants, chilling stress caused a more pronounced decrease in the rate of photosynthesis and stomatal conductance compared to plants grown in a complete nutrient solution. However, phosphorus resupply prevented photosynthesis inhibition in chilled plants and facilitated their recovery in terms of photosynthesis and stomatal conductance [111].

In conclusion, phosphorus deficiency has a significant impact on stomatal conductance and density in various plant species, leading to reduced photosynthetic performance, growth and water relations. Adequate P nutrition enhances plant resilience to abiotic stresses. Resupplying P to P-starved plants can prevent photosynthesis inhibition and improve the recovery of stomatal conductance and photosynthesis under abiotic stress conditions.

### 3.2. Biochemical Responses to Low P

Photosynthesis is influenced by a limited availability of P, as P plays a crucial role in adenosine triphosphate (ATP) production, which provides energy for this process [13]. The function of the electron transport chain, occurring in the thylakoid membranes of chloroplasts and responsible for ATP and NADPH production, can be affected by phosphorus deficiency.

The inadequacy of phosphorus can result in a decline in orthophosphate levels within the chloroplast stroma, consequently inhibiting the activity of ATP synthase [112]. In response to limited phosphorus availability, plants undergo adaptive changes to enhance photosynthetic efficiency. This adaptation involves a reduction in the number of photosystem II reaction centers and an augmentation in the activity of cyclic electron flow around photosystem I [113]. These adjustments enable plants to sustain ATP production despite the constrained phosphorus supply. Moreover, plants optimize respiratory efficiency under low phosphorus conditions by boosting the activity of alternative oxidase, which provides an alternate route for electron transport during respiration [113]. Such adaptations allow plants to uphold energy production and growth, even in the face of scarce phosphorus availability. Phosphorus plays a crucial role at various stages of the photosynthetic process, including ATP synthesis, the primary cellular energy source and the formation of ribulose-1,5-bisphosphate (RuBP), a vital molecule in the Calvin cycle. Insufficient phosphorus availability can impede ATP production, resulting in diminished photosynthetic activity and stunted plant growth. Numerous studies have demonstrated that limited phosphorus availability can reduce the photosynthetic rate in plants, resulting in diminished carbon fixation, plant growth and productivity [112,114,115]. For instance, a recent investigation explored the impact of low phosphorus availability on cotton genotypes with varying levels of tolerance. The study revealed that insufficient phosphorus decreased plant development, leaf area, gas-exchange parameters and antioxidant enzyme activity. Moreover, low phosphorus availability led to a decline in phosphoenolpyruvate carboxylase (PEPC) activity, ATP and NADPH content and steady chlorophyll a fluorescence [116].

### 3.3. Mesophyll Resistance under Low P

Apart from the direct effects on stomatal activity and photosynthetic units, phosphorus deficiency can also impact mesophyll resistance, which is a measure of CO_2_ diffusion from the stomatal pore to the photosynthesis site. Mesophyll resistance depends on several factors, including Rubisco concentration, an enzyme responsible for CO_2_ fixation during photosynthesis. Phosphorus deficiency in sugar beet plants decreased photosynthetic rates, increased mesophyll and diffusion resistances and reduced leaf phosphorus concentration. The decline in photosynthetic rates was correlated to increased stomatal diffusion resistance, which was greatly increased at low irradiance under low P [117]. It has been reported that smaller ATP content leads to a limitation in RuBP regeneration in P-deficient plants [118,119,120]. Additionally, the activity and amount of Rubisco are reduced under P deficiency, resulting in decreased net photosynthesis [118]. Moreover, the levels of chlorophyll and proteins in leaves may also be reduced by P deficiency [38]. In conclusion, P concentration plays a critical role in regulating plant growth and photosynthesis. P deficiency can reduce Rubisco content, increasing mesophyll resistance and reducing the CO_2_ fixation rate during photosynthesis [121,122]. This highlights the importance of proper phosphorus management in agriculture and the need for developing efficient and sustainable phosphorus management strategies.

## 4. Linking P Availability with Plant Tolerance to Environmental Stresses

Numerous genes encode transporters that aid in P acquisition from soil and then transport it to different parts of plants, including Golgi apparatus, chloroplast and mitochondria [2,123]. Here, we discuss critical genes important to Pi starvation and tolerance to abiotic stresses in plants (Table 1).

Novel candidate genes that are expressed under combined drought and Pi starvation stress have been reported [132,133]. These stresses influence the roots’ hydraulic property and aquaporin proteins, which regulate the transport of water. The expression of aquaporin genes is reduced in order to lower the synthesis of the apoplastic barrier [132]. In particular, the expression of *HvPIP2;2*, *HvPIP2;5*
*LcPIP1;1*, *LcPIP2;1* and *LcPIP2;4*, which are involved in the synthesis of aquaporins, is decreased. The reduced water conductivity of roots is also correlated to a decline in aquaporin expression. As a result, the ratio of the root-shoot surface is raised for further remobilization of resources to cope with stress conditions. Moreover, another study reported the expression of *HvPIP1; HvPIP2;2* and *HvPIP2;5* genes under drought and Pi deficiency. The synthesis of the apoplastic barrier considerably increased along the lateral roots and principal axis under low-nutrient-induced stress [133]. Further research is required to investigate which other gene families are involved in regulating the water movement during exposed combined stresses to plants.

*PHOSPHATE 1* (*PHO1*) is also from the Pi starvation family of genes implied in the triggering of signaling pathways and induces a molecular response to improve the *A. thaliana* productivity under Pi-exposed stress [125]. Recently, this gene was reported to play a critical role in interacting with various signaling pathways and making a stomata response to ABA in plants under salinity and drought stress. Most of the *PHO1* genes, i.e., *PHO1; H12/14; PHO1; H8* and *PHO1; H5*, were upregulated under high salinity stress (200 mM) in the roots of soybean genotypes [125]. However, the information on gene encoding for the transporting and signaling of Pi and crosstalk with other pathways and genes improved plant resilience against Pi starvation. Moreover, a de novo mediated molecular mechanism helped to explore the promising complex regulatory mechanisms for the development of sustainable agriculture practices that can help mitigate the negative impacts of environmental stress on plant growth and yield.

Climate change has resulted in the prevalence of abiotic stresses, which pose a significant challenge to plants due to their negative impact on physiological, metabolic, morphological and yield-related features. Abiotic stresses may occur in various forms, such as drought, heavy metal contamination, high- and low-temperatures, salinity and elevated CO_2_ levels, either singly or in combination, leading to significant damage to plants (Figure 4). Various agronomic practices have been employed to mitigate the effects of abiotic stresses, with fertilizer management being one of the most effective techniques to enhance plant growth and tolerance to extreme climatic conditions. In particular, the application of phosphorus (P) has been shown to be an efficient approach for improving plant tolerance to abiotic stresses (Table 2). Here we discuss the role of P in mitigating the impact of various abiotic stresses in plants.

### 4.1. Phosphorus and Drought Stress Tolerance

P application can mitigate the negative impact of drought and water deficit stress in plants. P fertilization improves plant characteristics including root and shoot length, biomass, water use efficiency, leaf water content, enzymatic activities, photosynthetic characters and metabolic profile, which enable the plant to grow under drought stress. An increase in the shoot and root biomass along with the yield has been reported to be improved by the application of P fertilizer in *Echinacea purpurea* [136], rapeseed [137] and *Phoebe zhennan* [140]. An improvement in the photosynthetic parameters, including leaf chlorophyll content, stomatal conductance, water use efficiency and quantum efficiency, of photosystem II due to the application of P fertilizers has been reported in rapeseed, *Alnus cremastogyne* [138], *Eucalyptus grandis* [139] and *Pisum sativum* [14]. P fertilization increases enzymatic activities including SOD, MDA, POD, CAT and organic osmolyte accumulation (e.g., proline) [14,138,140,157]. Additionally, proteins and metabolites related to drought tolerance are increased by P fertilization [12]. In addition to regulating phosphorus uptake and utilization, plants also have several mechanisms to cope with abiotic stress by modulating their phosphorus metabolism [12]. For example, when subjected to drought stress, plants may upregulate the expression of genes responsible for the synthesis and accumulation of compatible solutes, such as proline and glycine betaine, which help to protect the plant from the effects of water stress. These solutes can also aid in maintaining the turgor pressure of cells and prevent leaf wilting [158]. Shoot P accumulation and biomass are affected by root hair length, while the yield is affected only by root hair presence. Additionally, when combined with P deficits and drought stress, genotypes lacking root hairs are significantly less tolerant than those with root hairs [159]. Ca^2+^ and K^+^ -permeable ion channels are important elements that participate in plant tolerance to drought stress either by activating ABA-mediated stomatal regulations or activating the signaling cascades. Ion channels contribute to molecular and physiological mechanisms for stomatal closure, which prevent water loss under drought conditions [160]. Low phosphorus levels can affect the activity of ion channels such as Ca^2+^ and K^2+^. Low phosphorus levels lead to the adjustment of ion channel activity [161]. Ca^2+^-binding triggers changes in protein shape and charge. Similarly, phosphorylation imparts a negative charge, altering protein conformations and their interactions [162]. Phosphorus supply also enables plants to alleviate drought stress by downregulating hormonal biosynthesis and ROS generation [163,164]. The application of P under drought stress upregulated the genes responsible for aquaporins channels, which are responsible for facilitating the transit of water and small solutes through cell membranes [132].

### 4.2. Phosphorus and Salinity Stress Tolerance

Salinity stress induces an osmotic and ionic imbalance in plants and leads to a decrease in plant growth and yield. P application showed promising effects on alleviating the negative impacts of salinity stress in plants. P fertilization improved growth parameters including root and shoot length, leaf hair and trichome densities and yield in quinoa [143], *Aeluropus littoralis* [145], sugar beet [144] and okra [149]. Higher water and nutrient uptake, photosynthetic characteristics including leaf chlorophyll content, stomatal conductance and water use efficiency have been improved by P fertilization in different crops [142,146,147]. Low phosphorus fertilization can alleviate the negative impacts of salt stress on wheat plants, leading to improved photosynthetic performance, nutrient uptake and growth, highlighting the importance of optimizing phosphorus availability to mitigate the detrimental effects of salinity [142]. Low phosphorus availability enhances salt tolerance in maize plants by promoting changes in root and shoot growth, tissue density, osmolyte accumulation and ion selectivity, suggesting that phosphorus deficiency modifies the physiological mechanisms underlying salt stress response in maize [146]. Higher activities of enzymes, proline accumulation and sugar content were positively affected by P fertilization [144,145,148]. Arbuscular mycorrhizal fungi (AMF) and leaf phosphorus (Pi) supplementation improved photosynthesis and biomass and reduced sodium and chloride accumulation under saline conditions [147]. Furthermore, plants have developed several mechanisms to tolerate high salinity by modulating their phosphorus metabolism. One of the key mechanisms is the accumulation of high levels of polyP in cells. These polyP molecules act as a buffer against changes in the intracellular pH and they also play a role in the transport and storage of phosphate [165,166]. The exclusion of Na^+^ ions from the cell to the extracellular area is one of the major physiological mechanisms of plants to alleviate the negative impact of salinity stress. Phosphorus fertilizer led to lower Na^+^ concentration in roots and provided tolerance to salinity stress [143]. Phosphorus fertilization also mitigated the adverse effects of salinity on quinoa plants by improving nutrient availability, reducing the Na^+^/K^+^ ratio and maintaining the balances of other essential minerals, such as Mg^2+^, Ca^2+^ and Fe^2+^, while preventing a decrease in leaf dry matter. Exogenous supplication of phosphorus improved salt tolerance by increasing Na^+^ exclusion [167].

### 4.3. Phosphorus and Temperature Stress Tolerance

P fertilization has shown its promising impact on alleviating the negative effects of high- and low-temperature stress in plants. Application of P fertilization in rice, wheat and soybean improved low- and high-temperature stress tolerance [151,152,153]. P fertilization showed positive effects on grain yield and grain quality in wheat and rice under high- and low-temperature stress [151,153]. Photosynthetic characteristics such as stomatal conductance with higher dry matter accumulation, transportation of accumulation, partitioning and higher yield have been increased due to P fertilization [151,152]. Stomatal regulations play critical roles in mitigating heat stress to plants by increasing the transpiration rate and lowering the leaf microclimatic temperature. Phosphorus fertilization, when combined with an optimal level of nitrogen, enhanced the physiological traits, seed yield and oil yield of rapeseed, suggesting its potential role in improving heat stress tolerance by increasing gas exchange and the quantum yield of photosystem II, while nitrogen application alone at higher levels may not provide significant additional benefits [168]. Modulation of photosynthesis, stomatal conductance and water use efficiency are the most common underlying mechanisms to alleviate the negative impacts of heat stress. A study carried out by Fahad et al. showed the application of phosphorus combined with biochar improved heat stress tolerance by improving the stomatal conductance, water use efficiency and grain size, which showed the impact of phosphorus on stomatal regulations under heat stress [169]. The application of P also plays a role in mitigating the negative impact of low-temperature stress by modulating photosynthesis and improving the accumulation and partitioning of dry matter [170].

### 4.4. Phosphorus and Heavy Metal Stress Tolerance

Heavy metal stress including Hg, Cu, Ni and Al led to changes in plant physiological and molecular processes, which resulted in stunted growth and, ultimately, plant death. Plants apply a diverse range of adaptive strategies to alleviate the negative and toxic impacts of heavy metals. P supplementation enhanced plant tolerance to heavy metals by influencing their physiological and metabolic processes [163]. Studies by Huang et al. and Wu et al. demonstrated that P fertilization promotes plant growth, increases shoot and root biomass and enhances enzymatic activities, thereby supporting plant growth and development [154,171]. Additionally, P fertilization positively affects aluminum (Al) tolerance in Lespedeza bicolor by reducing Al uptake and accumulation, while increasing P accumulation and translocation from roots to shoots [155]. Plants also have several mechanisms to tolerate heavy metal toxicity by modulating their phosphorus metabolism. For example, plants can reduce the uptake of heavy metals by reducing the expression of metal transporters [172]. Additionally, some plants have developed the ability to accumulate high levels of phytochelatins, which are small peptides that bind to heavy metals and help to detoxify them [173]. Application of P decreased the uptake and accumulation of Cd and Pb significantly in the plants treated with P fertilizers in maize, rapeseed and soybean [163]. Another study conducted by Cao et al. showed the reduction of Pb uptake in shoots and roots through the use of P fertilization [174], which was supported by increasing the SOD and other enzymatic activities [175]. Overall, the application of P shows different impacts on different heavy metals and further research is needed.

### 4.5. Phosphorus, Waterlogging and Elevated CO_2_ Stress Tolerance

Under waterlogging stress, P availability is higher compared to the normal moisture content due to the formation of aerenchyma tissues. Application of P fertilization showed an increase in grain yield in fodder maize under waterlogging conditions [156,176]. The effects of phosphorus deficiency and waterlogging stress on plant growth can be significant. It is possible for plants to become nutrient deficient when they are subjected to waterlogging stress. Researchers have found that waterlogged seedlings of various species have reduced foliar nutrients, with significantly reduced concentrations of nitrogen, phosphorus and potassium [177]. A combination of soil waterlogging and P depletion will also affect the relative growth rate and survival of plants. However, when soil phosphorus is supplied above adequate levels, it appears that there is a substantial increase in the number and size of xylem vessels, which can facilitate the plant’s tolerance to waterlogging stress and may result in an increase in the potential root hydraulic conductance per leaf area. It is essential to have adequate phosphorus and drainage for optimal plant growth under waterlogging conditions [178]. There are very few reports regarding the application of P in terms of improving the plant’s tolerance to waterlogging stress, and the underlying mechanism is yet to be discovered. An increased CO_2_ concentration changes plant morphology and physiological characteristics in a negative manner. The increasing atmospheric CO_2_ concentration has implications for nutrient cycling, particularly P dynamics in ecosystems. Understanding the effects of elevated CO_2_ on plant P demand, utilization and acquisition from soil is crucial for sustainable P management in P-deficient regions. Elevated CO_2_ is expected to lead to significant increases in plant P demand due to stimulated photosynthesis and subsequent growth responses. Changes in root morphology, increased rooting depth and alterations in root exudate quantity and composition have been observed under elevated CO_2_, potentially impacting P acquisition through the chelation of P, biochemical environment modifications and changes in microbial activity in the rhizosphere. Further research is needed to explore the chemical, molecular, microbiological and physiological aspects of how elevated CO_2_ influences plant P utilization and acquisition [179].

## 5. Future Prospects

The biological availability of phosphorus for plants is restricted, as it tends to get fixed, absorbed or precipitated within the soil. Moreover, when plants encounter adverse environmental conditions like drought or salinity, they confront even greater obstacles in obtaining and effectively utilizing phosphorus. To address this issue, various phosphorus-based interventions have been developed to enhance P use efficiency in plants. A detailed comparison of the most commonly employed agronomic strategies are listed below in Table 3.

The development and application of phosphorus-based interventions are crucial in meeting the growing demand for food production while minimizing environmental impacts. Although these interventions help plants to adapt to adverse environmental conditions, we need more sustainable solutions based on the understanding of the underlying molecular mechanisms to cope with the changing climatic conditions in the future. For this purpose, the following research directions can be critical:▪It is essential to delve deeper into the relationship between phosphorus and abiotic stresses considering the bidirectional impact they have on each other. While phosphorus aids plants in adapting to stress conditions, it is crucial to recognize that stresses can also affect a plant’s ability to acquire phosphorus, forming a negative feedback loop. To overcome this challenge, it is necessary to focus on identifying and understanding the mechanisms underlying phosphorus tolerance in plants, particularly in the context of abiotic stresses. Investigating the genetic and physiological basis of phosphorus tolerance and exploring the molecular pathways involved will pave the way for developing crop varieties with enhanced phosphorus tolerance and stress resilience.▪Phosphorus use efficiency (PUE) is a critical factor that can be influenced by abiotic stresses. By improving PUE, we can not only address phosphorus-related issues in agriculture but also enhance plant resilience to abiotic stresses. Therefore, it is crucial to investigate and interpret the mechanisms governing phosphorus use efficiency in plants under stress conditions. Understanding the molecular and physiological processes that contribute to efficient phosphorus utilization will provide insights into how to develop innovative strategies for improving PUE in crops. Moreover, identifying and characterizing crop varieties with high phosphorus use efficiency that can withstand abiotic stresses should be a focal point for future breeding programs.▪Given the close relationship between phosphorus, stomata and abiotic stresses, it is imperative to explore the specific interactions and mechanisms involved in the “P-stomata-Abiotic stresses” nexus. The stomata, as key regulators of water loss and gas exchange, have a profound impact on plant tolerance to abiotic stresses such as drought, salinity and high temperatures. Integrating the role of phosphorus in stomatal regulation under stress conditions can provide valuable insights into how to improve plant resilience and stress tolerance.▪Excessive phosphorus (P) can also elicit diverse and complex responses in plants, impacting both molecular and physiological modifications. At the molecular level, it alters gene expression, signaling pathways and epigenetic modifications. Physiologically, P promotes enhanced growth, increased photosynthesis and changes in root architecture. However, excess P can also disrupt nutrient balances, trigger oxidative stress and affect mycorrhizal associations. Understanding these responses is crucial for optimizing phosphorus management in agriculture and ensuring sustainable plant growth under high P use efficiency.

By exploring these research directions, we can greatly enhance our understanding of the complex connection between phosphorus, stomata and abiotic stresses. This knowledge will pave the way for precise strategies to counteract the detrimental effects of stresses on phosphorus absorption and usage. Ultimately, this will enable us to develop crop varieties that yield higher outputs thanks to their improved ability to tolerate phosphorus, utilize it efficiently, maintain optimal stomatal functioning and withstand various environmental stresses. These advancements hold immense promise, not only in promoting sustainable agriculture but also in tackling the urgent challenge of global food security, particularly in the face of unpredictable climatic conditions.

## 6. Conclusions

This review underscores the vital role of phosphorus as a central regulator in eliciting a plant’s physiological responses to abiotic stresses, shedding light on its multifaceted influence on critical processes such as root system architecture, nutrient uptake, photosynthesis and stomatal regulation. By orchestrating these intricate mechanisms, phosphorus emerges as a key player in bolstering plant resilience against adverse environmental conditions. Moreover, the significance of stomata in mediating the intricate interplay between phosphorus availability and abiotic stresses is unequivocally emphasized. Serving as pivotal gatekeepers, stomata meticulously regulate water loss and gas exchange, thereby assuming a paramount role in facilitating plant adaptation to challenging environments. Recognizing the pivotal role of stomata in the context of phosphorus dynamics and abiotic stress responses is indispensable for devising efficacious strategies to enhance plant performance and fortify stress tolerance. By unraveling the intricate molecular pathways involved in phosphorus signaling and stomatal regulation, scientists can unlock novel avenues for crop enhancement, ushering in a new era of high-yielding crop varieties fortified with exceptional phosphorus tolerance, increased phosphorus use efficiency and unparalleled abiotic stress resilience. These transformative advancements not only propel sustainable agriculture forward but also address the pressing global concern of ensuring food security amidst the backdrop of rapidly changing climatic conditions. Therefore, further research in this area is warranted to fully exploit the potential of phosphorus and stomata in enhancing plant resilience to abiotic stresses and ensuring global food production sustainability.

## Figures and Tables

**Figure 1 plants-12-02861-f001:**
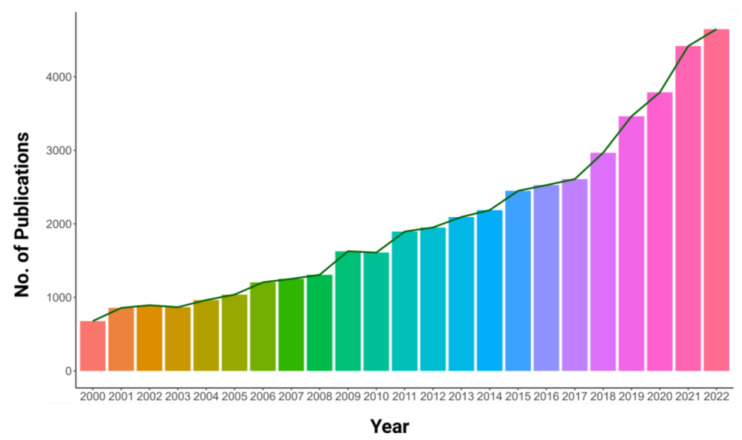
Publications per year based on Scopus data from 2000 to 2022, quantifying “phosphorus” studies in “plants”.

**Figure 2 plants-12-02861-f002:**
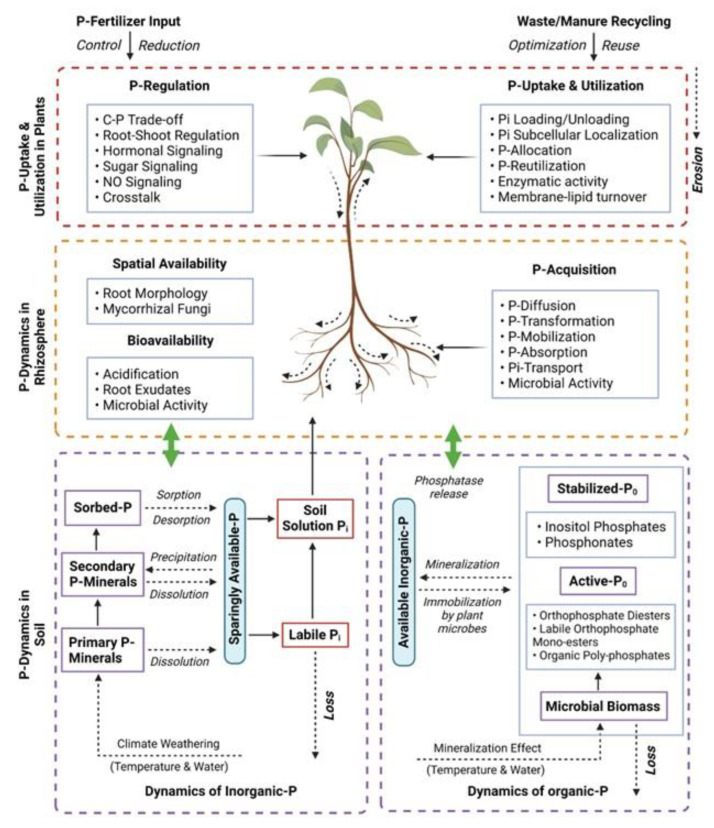
A diagrammatic representation of phosphorus dynamics from soil to plants. Adapted from [25] with modifications. Created using BioRender (accessed on 23 March 2023, https://biorender.com/).

**Figure 3 plants-12-02861-f003:**
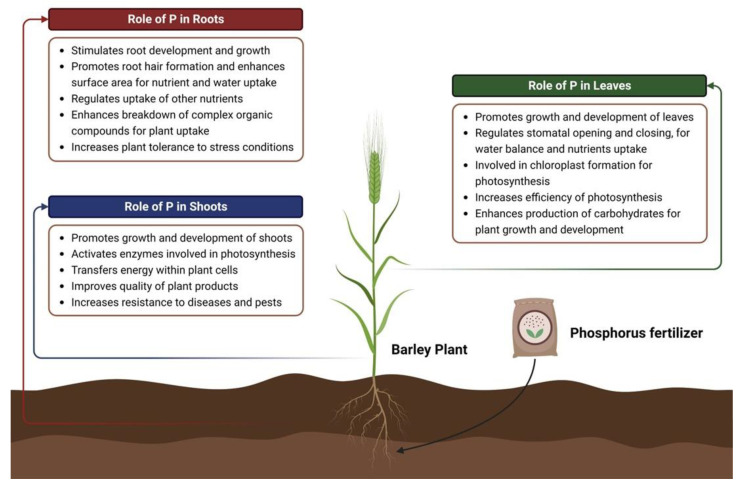
Localized functions of phosphorus in different parts of the plant. Created using BioRender (accessed on 4 May 2023, https://biorender.com/).

**Figure 4 plants-12-02861-f004:**
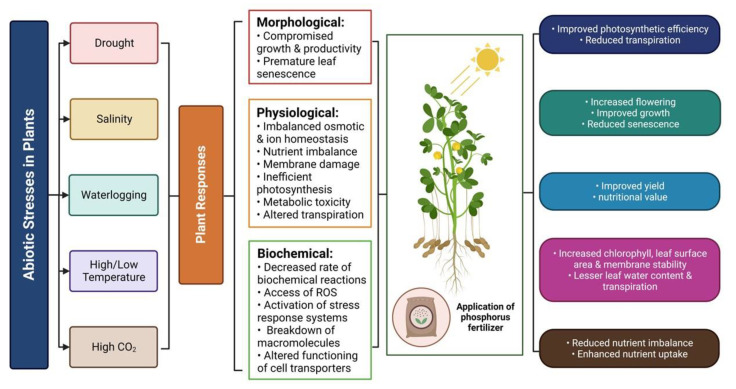
Role of phosphorus in mitigating the impact of abiotic stresses in plants. Created using BioRender (accessed on 23 March 2023, https://biorender.com/).

**Table 1 plants-12-02861-t001:** Genes and their corresponding functions involved in P starvation or combined abiotic stress tolerance.

Plants	Potential Gene	Family	Abiotic Stress Tolerance	Functions	Reference
**Tomato**	*Lept1 and lept2*	*Lept*	Low Pi	Regulation of P homeostasis	[124]
** *Glycine max* **	*H12/14 H1/4, PHO1; H5 and H8*	*PHOSPHATE1 (PHO1)*	Low Pi and salt stress	Tolerance against salt stress Morphological adaptation and divergence	[125]
	*GmETO1*	*Ethylene Overproduction Protein*	Low Pi	Development of root under Pi starvation and expressed notably for ethylene biosynthesis	[126]
	*GmPAP12*	*GmPAP*	Low Pi	Stimulate the purple acid phosphatases synthesis for nodules to absorb more Pi	[127]
	*GmSPX-RING1*	*SPX-RING*	Low Pi	Control the efficiency of Pi in different soybean cultivars	[128]
	*GmPHR25*	*GmPT Pi transporter*	Low Pi	Upregulated in root hairs to facilitate the absorption of Pi	[129]
** *Arabidopsis thaliana* **	*microRNA399* and *PHR1*	*PHR1*	Low Pi	Regulation of P homeostasis	[130]
** *Cerasus pseudocerasus* **	*CpERF7*	*ERF*	Drought and low Pi	Stimulate the auxin-mediated stress-responsive genes to increase tolerance against drought and low P	[131]
** *Hordeum vulgare* **	*HvPIP2;1; HvPIP2;2, HvPIP1;3, HvPIP2;4 and HvPIP2;5*	*Plasma membrane intrinsic proteins*	Drought and low Pi	Encode water channels that facilitate the transit of small solutes and water through cell membrane	[132,133]
	*HvPHT1;1, HvPHT1;3, HvPHT1;4, HvPHT1;6,*	*PHT*	Arsenic stress	Reduced the toxicity of arsenic stress	[123]
**Tobacco**	*NtPIN3*	*Cytoplasmic membrane transporter*	Low Pi	Increased tolerance against low Pi by modifying root elongation	[134]
** *Brassica napus* **	*PHO2*	*PHO*	Low Pi	Enhance the uptake and transportation of Pi in soil	[135]
** *Leymus chinensis* **	*LcPIP1;1 LcPIP2;1, LcPIP2;4;*	*Plasma membrane intrinsic proteins*	Low Pi and drought stress	PIP genes are overexpressed in drought or Pi starvation, which resulted in alleviation of abiotic stresses	[132]

**Table 2 plants-12-02861-t002:** Reported effect of phosphorus on plant tolerance to drought, salinity, temperature, high CO_2_ and heavy metal stresses.

Abiotic Stresses	Plant	P application	Alleviation Mechanisms	References
**Drought**	*Echinacea purpurea*	Split plot experiment was performed with drought stress in term of available water depletion (25, 50 and 75%) to retain the filed capacity. P fertilizer was used as control without P fertilizer, 100% plant required P from triple super-phosphate, sole application of mycorrhizal arbuscular fungi (AMF) and *Pseudomonas fluorescens* bacteria (PFB) and AMF or PFB combination with 50% of the plant’s P requirement.	Application of P increased root biomass and yield by 57% and 47%, respectively, under drought conditions. In 25% of AWD, the highest root cynarin (0.583 mg/g dry matter) was observed in the joint application of phosphorus + AMF. Together with the AMF, P improved the drought tolerance traits.	[136]
**Water deficit**	Rapeseed	Plants were grown under different irrigation levels (70-, 100-, 130- and 160-mm evaporation from class A pan, respectively), which were treated with five fertilizers (control, chemical fertilizer including N and P (about 300 and 150 kg ha^−1^, respectively, based on soil analysis)).	P fertilizers especially in combination with other fertilizers decreased proline content and leaf temperature, with an increase in antioxidants and enzymatic activities including chlorophyll content, leaf water content, membrane stability index and stomatal conductance with improved yield.	[137]
**Drought**	*Alnus cremastogyne*	P fertilizer was applied in the form of NaH_2_PO_4_ (25.5% P) to each of the pots in each 30-day interval 3 times in the entire crop duration.	Application of P improved the relative water content, photosynthesis rate, increase in antioxidative enzymes including SOD, CAT, POD, osmolytes accumulation, soluble proteins and decrease in lipid peroxidation levels.	[138]
**Drought**	*Eucalyptus grandis*	P fertilizer was applied in the form of NaH_2_PO_4_ (25.5% P) to each of the pots in each 30-day interval 3 times in the entire crop duration.	P application alleviated the drought stress effect by improving the leaf relative water content, net photosynthesis, quantum efficiency of photosystem II and amelioration in some other physiological traits related to drought stress tolerance.	[139]
**Drought**	*Phoebe zhennan*	P fertilizer was applied in the form of NaH_2_PO_4_ (25.5% P) to each of the pots in each 30-day interval.	P alleviated the drought effect by increasing the relative water content, net photosynthesis rate, higher quantum efficiency, higher rooting systems, enhanced root biomass, decrease in MDA and upregulations of photosynthetic pigments, osmolytes and nitrogenous compounds.	[140]
**Drought**	*Pisum sativum*	P was applied in the form of KH_2_PO_4_ at two different rates: 15 (P15) and 60 mg P kg^−1^ (P60), mixed evenly throughout the soil.	Optimal P application enhanced the water use efficiency, soluble sugars and relative water content. Application of P increased the salt tolerance index with higher root length, nodule number, stomatal conductance and uptake of nitrogen.	[14]
**Drought**	Soybean	The P levels (added as KH_2_PO_4_) were 0, 15 and 30 mg P kg^−1^ soil.	Addition of P enhanced the concentration and accumulation of nitrogen in shoots and seeds. P application mitigated the drought effects on plant by increasing the protein concentration.	[141]
**Salinity**	Wheat	P was used in the form of Ortho P fertilizer, phosphoric acid-based fertilizers with K and N containing 52% and 62% of P2O5, respectively, with 100% orthoP for each one.	Sufficient P application alleviated the salinity stress by promoting growth and reducing salt toxicity. Fertilized plants have higher shoot and root dry weights under salinity stress.	[142]
**Salinity**	Quinoa	Filed experiment was set up with different EC levels of irrigation water (5, 12 and 17 dS·m^−1^). P was added in the form of P2O5 at the rate of 0, 60 and 70 kg of P_2_O_5_ ha^−1^.	P application under saline conditions minimized the effect of salinity and improved the yield with higher water and nutrient uptake. The results suggested that P application minimizes the adverse effects of high soil salinity and can be adopted as a coping strategy under saline conditions.	[143]
**Salinity**	Sugar beet	Plants were grown under salinity water with different EC values of 0.7, 4, 8 and 12 dS·m^−1^). P was applied in the form of P2O5 with a rate of 100, 120 and 140 kg P_2_O_5_·ha^−1^.	P improved the rate of yield and sugar content of sugar beets under the tested salinity levels.	[144]
**Salinity**	*Aeluropus littoralis*	Plants were grown at moderate salinity (100 mM NaCl). P fertilizers were applied at different doses: low, moderate and high (5 mM, 60 mM and 180 mM KH_2_PO_4_).	P fertilization improved the salinity tolerant characteristics including increase in leaf hair and trichome densities, total polyphenol content and total antioxidant capacity in plants cultivated.	[145]
**Salinity**	Maize	P was introduced to the nutrient medium in the form of KH_2_PO_4_, with concentrations of 5 μM considered as low P and 200 μM as high P.	Higher P application prevented leaf chlorosis under salinity stress with improved	[146]
**Salinity**	Woody species	Plants were grown in a factorial 2 × 2 × 2 (presence or absence of salt, AMF, or P_i_), totaling eight treatments: control, P_i_, AMF, AMF ∗ P_i_, salt, salt ∗ P_i_, salt ∗ AMF and salt ∗ AMF ∗ P_i_, with eight replicates each and one plant per pot, totaling 64 experimental units.	P fertilization increased biomass and photosynthetic pigments under salinity conditions. Metabolites were also positively impacted due to fertilization.	[147]
**Salinity**	*Phaseolus vulgaris*	Plants were grown under different salinity conditions (1.56, 4.78 and 8.83 dS·m^−1^) and P fertilizer at the rate of 0, 30, 60 and 90 kg ha^−1^.	P application significantly increased the total chlorophyll content, total soluble sugars, carotenoids, total free amino acid and proline with higher accumulation of K^+^, Ca^2+^, Mg^2+^ and higher tolerance and yield.	[148]
**Salinity**	Okra	Plants were grown under two different salinity conditions (4 and 8 dS·m^−1^) and combined with different rates of P fertilizers (0, 30, 60 and 90 kg ha^−1^ from triple super phosphate).	Application of P increased the green and dry pod yield under salinity stress.	[149]
**Salinity**	*Cicer arietinum* L.	Plants were grown under two salinity conditions (0 and 150 mM NaCl) and treated with three P fertilizers (0, 90, 200 kg h^−1^ of P_2_O_5_ in the form of super triple phosphate).	P application improved plant growth. P fertilization increased the leghemoglobin (92%), reduced proline content (−69%) and protected membranes against peroxidation compared to saline conditions. Also, yield was increased due to the P fertilizers.	[150]
**Low temperatures**	Wheat	Pot experiment was conducted with tolerant and sensitive cultivar and was treated under chilling (T1 at 4 °C) and freezing treatment (T2 at −4 °C) as well as ambient temperature (CK at 11 °C) during the anther differentiation treated with P fertilizers.	Application of P alleviated low-temperature stress by increasing the stomatal conductance, dry matter accumulation, transportation of assimilates, grain number per spike, 1000 grain weights, yield per plant.	[151]
**Low and high temperatures**	Soybean	Plants were grown at different temperatures (22, 26, 30 and 34 °C corresponds to moderately low, optimum, moderately high and high temperature) with combination of 0.5 mM and 0.08 mM P nutrition.	Sufficient P fertilization improved the temperature stress tolerance by increasing the plant efficiency in utilization and biomass partitioning to pods.	[152]
**High-temperature stress**	Rice	Plants were grown in controlled growth conditions with high day and night temperatures (35 °C ± 2 and 32 °C ± 2, respectively). P fertilizers were added singly or combined with biochar.	P fertilization ameliorated the adverse impact of high temperatures with higher grain formation and grain quality.	[153]
**Heavy metals**	-	Five different types of treatments were added: control (C), heavy metal pollution (H), heavy metal pollution + nitrogen (HN), heavy metal pollution + phosphorus (HP), heavy metal pollution + nitrogen and phosphorus (HNP)	P addition of HP and HNP treatments restored plant species richness and increased plant diversity under heavy metal pollution. P addition had a better performance in restoring the species composition and relative dominance of plant communities.	[15]
**Heavy metals**	Maize	Maize plants were grown in soil collected from paddy field located near recycle area and fertilizers were applied as two amendments (calcium–magnesium phosphate fertilizer, PF; commercial organic fertilizer, OF) tested in single and in combination.	Application of P increased the maize shoot and root biomass and enzymatic activities such as urease and catalase. Additionally, microbial community structure was improved.	[154]
**Heavy metals**	*Lespedeza bicolor; Lespedeza cuneata*	Two species, *Lespedeza bicolor* and *L. cuneata*, were grown for 30 d with alternate Al and P treatments in a hydroponics system.	P application improved root growth and heavy metal tolerance by lowering the Al uptake and accumulation. Enhancement of Al resistance by P in the resistant species might be associated with its more efficient P accumulation and translocation to shoots and greater Al exclusion from root tips after P application.	[155]
**Waterlogging**	*Brachlaria grass*	Tolerant and sensitive cultivars were grown under control and waterlogged conditions with two different fertilizer levels (low and high).	Higher availability of P under waterlogged soil imparted the tolerance.	[156]
**Drought and elevated CO_2_**	Field pea (*Pisum sativum*)	Plants were grown in P-deficient vertisol, supplied with two doses of P (15 mg and 60 mg Kg^−1^) and treated with ambient and elevated CO_2_ (380–400 ppm and 550–580 ppm).	P improved water use efficiency under elevated CO_2_ levels.	[14]

**Table 3 plants-12-02861-t003:** Strategies, description, mode of actions, benefits and limitations of available P-based interventions to enhance P use efficiency under abiotic stress conditions in plants.

Strategies	Description	End Products	Mode of Action	Benefits	Limitations
**Chemical fertilizers**	These are synthetic fertilizers that contain a high concentration of P in the form of water-soluble salts. They are widely used in agriculture to provide quick and efficient nutrition to plants	Biomass, increased yield	Direct supply of P to plants, quick response	Quick and efficient nutrient supply, high P concentration	Expensive, environmental pollution, soil degradation, reduced microbial activity
**Organic amendments**	These are natural sources of P such as animal manure, compost and bone meal. They provide a slow and steady release of P through the breakdown of organic matter in the soil	Biomass, improved soil structure, microbial activity	Slow release of P through mineralization, increased soil fertility	Slow and steady release of P, improved soil structure and health, increased microbial activity	Requires large quantities for adequate P supply, potential for nutrient imbalances, may contain pathogens or weed seeds
**Biofertilizers**	These are microbial inoculants that contain P-solubilizing microorganisms such as bacteria, fungi and algae. They enhance the availability of P in the soil and improve nutrient uptake by plants	Biomass, improved soil health	Biological fixation of atmospheric P into plant-available form, symbiotic relationship with plants	Improved nutrient uptake and stress tolerance, reduced environmental pollution, improved soil health	Limited availability and variability, potential for ineffective inoculants
**Nano P fertilizers**	These are engineered nanoparticles that contain P and are designed to improve the solubility and bioavailability of P in the soil. They are considered to be highly efficient and cost-effective	Improved nutrient uptake, increased yield	Enhanced solubility and bioavailability, reduced leaching and environmental impact	Highly efficient and cost-effective, improved solubility and bioavailability of P, reduced environmental pollution	Limited research on potential environmental and health impacts
**Mycorrhizal inoculants**	These are symbiotic associations between plants and fungi that enhance the absorption of P from the soil. They also improve plant growth and nutrient uptake under stress conditions	Improved nutrient uptake, increased resistance to stress	Enhanced surface area and absorption, improved soil structure and nutrient cycling	Improved P uptake and nutrient use efficiency, enhanced plant growth and stress tolerance	Limited host range, potential for ineffective inoculants
**Genetic engineering**	This involves the manipulation of genes involved in P transport, signaling and metabolism in plants to enhance their efficiency in utilizing P	Improved nutrient uptake and utilization	Manipulation of genes involved in P transport, signaling and metabolism	Improved P use efficiency and stress tolerance, enhanced nutrient uptake, potential for long-term sustainability	Limited research on environmental and health impacts, potential for unintended consequences
**Plant growth-promoting rhizobacteria**	These are soil bacteria that colonize the root zone of plants and improve their growth and nutrient uptake by producing phytohormones and metabolites	Increased growth, improved nutrient uptake and tolerance to abiotic stress	Production of phytohormones and metabolites, biocontrol of plant pathogens	Improved nutrient uptake and stress tolerance, enhanced plant growth, reduced environmental pollution	Limited effectiveness on some plant species, potential for ineffective inoculants
**Biostimulants**	These are natural or synthetic substances that enhance plant growth and nutrient uptake by stimulating physiological processes in plants	Increased growth, improved stress tolerance	Activation of physiological and biochemical pathways, enhanced nutrient uptake and utilization	Improved plant growth and nutrient uptake, enhanced stress tolerance, reduced environmental pollution	Limited research on effectiveness and potential environmental impacts
**P-solubilizing microorganisms**	These are microorganisms that enhance the solubility and availability of P in the soil by releasing organic acids and enzymes that break down P-containing minerals	Increased P availability, improved soil health	Solubilization and mineralization of soil P, biological fixation of atmospheric P	Improved solubility and bioavailability of P, reduced environmental pollution, improved soil health	Limited research on effectiveness and potential environmental impacts
**Soil acidification**	This involves the application of acidic substances such as sulfur or ammonium sulfate to lower the pH of the soil, thereby increasing the solubility and bioavailability of P	Improved P availability, increased yield	Lowered pH increases solubility and bioavailability of soil P	Improved solubility and bioavailability of P, reduced environmental pollution	Potential for soil degradation, increased leaching of other nutrients, potential for negative impacts on soil microorganisms

## Data Availability

Not applicable.

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
