# Peer review of "Phosphorus Plays Key Roles in Regulating Plants’ Physiological Responses to Abiotic Stresses"

_plants, 2023, doi:10.3390/plants12152861_

Round 1
Reviewer 1 Report
Please, see the attached file.

I would recommend that the text be reviewed by a professional English language editor because there are quite a few minor linguistic inaccuracies that would make it difficult for readers.
Reviewer 2 Report
The manuscript is well written and the topic also is informative about phosphorus participates in the physiological responses of plants to abiotic stresses. This study is well-organized and interesting. However, a few important points must be taken cares which are indicated below:
1. Compared to low-P, excess P can also trigger complex and diverse responses in plants, and this should be discussed in the manuscript.
2. Is there a link between phosphorus responses and hormone signaling pathways in different plants? Plants have developed various molecular mechanisms to cope with P phosphorus deficiency and excess by modifying different hormone signaling pathways, and many TFs were involve in these responses.
N/A
Round 2
Reviewer 1 Report
The authors have added missing elements from the exhibition.
Again, the format for arranging the chapters that an article published in the journal Plants should contain is not followed.
The Materials and Methods chapter should contain the databases and search engines that were used.
I will not argue with the authors, this is a journal regulation that we all follow both when writing and reviewing articles.
Author Response
Thanks for your efforts in reviewing our manuscript and your endorsement of our work. Given that this is a review paper, we decide not to have the 'materials and method' in our manuscript. We hope you can understand.
Reviewer 2 Report
Thanks for the author's serious response. This work meets the requirements of this journal and can be published.
Author Response
Thanks for your endorsement to our work!